# Space-Time Correspondence as a Contrastive Random Walk

**Allan A. Jabri**
UC Berkeley

**Andrew Owens**
University of Michigan

**Alexei A. Efros**
UC Berkeley

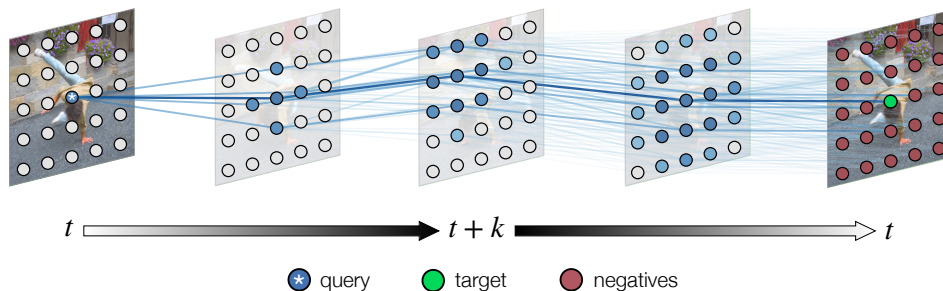

query    target    negatives

Figure 1: We represent video as a graph, where nodes are image patches, and edges are affinities (in some feature space) between nodes of neighboring frames. Our aim is to learn features such that temporal correspondences are represented by strong edges. We learn to find paths through the graph by performing a random walk between query and target nodes. A contrastive loss encourages paths that reach the target, implicitly supervising latent correspondence along the path. Learning proceeds *without labels* by training on a *palindrome* sequence, walking from frame $t$ to $t + k$, then back to $t$, using the initial node itself as the target. Please see our webpage for videos.

## Abstract

This paper proposes a simple self-supervised approach for learning a representation for visual correspondence from raw video. We cast correspondence as prediction of links in a space-time graph constructed from video. In this graph, the nodes are patches sampled from each frame, and nodes adjacent in time can share a directed edge. We learn a representation in which pairwise similarity defines transition probability of a random walk, such that prediction of long-range correspondence is computed as a walk along the graph. We optimize the representation to place high probability along paths of similarity. Targets for learning are formed without supervision, by cycle-consistency: the objective is to maximize the likelihood of returning to the initial node when walking along a graph constructed from a palindrome of frames. Thus, a single path-level constraint implicitly supervises chains of intermediate comparisons. When used as a similarity metric without adaptation, the learned representation outperforms the self-supervised state-of-the-art on label propagation tasks involving objects, semantic parts, and pose. Moreover, we demonstrate that a technique we call edge dropout, as well as self-supervised adaptation at test-time, further improve transfer for object-centric correspondence.

## 1 Introduction

There has been a flurry of advances in self-supervised representation learning from still images, yet this has not translated into commensurate advances in learning from video. Video is often treated as a simple extension of an image into time, modeled as a spatio-temporal $XYT$ volume [71, 117, 14]. Yet, treating time as yet another dimension is limiting [26]. One practical issue is the sampling rate

mismatch between $X$ and $Y$ vs. $T$. But a more fundamental problem is that a physical point depicted at position $(x, y)$ in frame $t$ might not have any relation to what we find at that same $(x, y)$ in frame $t + k$, as the object or the camera will have moved in arbitrary (albeit smooth) ways. This is why the notion of *temporal correspondence* — "what went where" [112] — is so fundamental for learning about objects in dynamic scenes, and how they inevitably change.

Recent approaches for self-supervised representation learning, such as those based on similarity learning [17, 22, 99, 87, 113, 44, 95, 40, 16], are highly effective when pairs of matching views are assumed to be known, e.g. constructed via data augmentation. Temporal correspondences, however, are *latent*, leading to a chicken-and-egg problem: we need correspondences to train our model, yet we rely on our model to find these correspondences. An emerging line of work aims to address this problem by bootstrapping an initially random representation to infer which correspondences should be learned in a self-supervised manner e.g. via cycle-consistency of time [109, 105, 57]. While this is a promising direction, current methods rely on complex and greedy tracking that may lead to local optima, especially when applied recurrently in time.

In this paper, we learn to associate features across space and time by formulating correspondence as pathfinding on a space-time graph. The graph is constructed from a video, where nodes are image patches and only nodes in neighboring frames share an edge. The strength of the edge is determined by similarity under a learned representation, whose aim is to place weight along paths linking visually corresponding patches (see Figure 1). Learning the representation amounts to fitting the transition probabilities of a walker stepping through time along the graph, reminiscent of the classic work of Meila and Shi [67] on learning graph affinities with a local random walk. This learning problem requires supervision — namely, the target that the walker should reach. In lieu of ground truth labels, we use the idea of cycle-consistency [121, 109, 105], by turning training videos into *palindromes*, e.g. sequences where the first half is repeated backwards. This provides every walker with a target — returning to its starting point. Under this formulation, we can view each step of the walk as a contrastive learning problem [17], where the walker's target provides supervision for entire chains of intermediate comparisons.

The central benefit of the proposed model is efficient consideration and supervision of many paths through the graph by computing the expected outcome of a random walk. This lets us obtain a learning signal from all views (patches) in the video simultaneously, and handling ambiguity in order to learn from harder examples encountered during training. Despite its simplicity, the method learns a representation that is effective for a variety of correspondence tasks. When used as a similarity metric without any adaptation, the representation outperforms state-of-the-art self-supervised methods on video object segmentation, pose keypoint propagation, and semantic part propagation. The model scales and improves in performance as the length of walks used for training increases. We also show several extensions of the model that further improve the quality of object segmentation, including an edge dropout [91] technique that encourages the model to group "common-fate" [111] nodes together, as well as test-time adaptation.

## 2   Contrastive Random Walks on Video

We represent each video as a directed graph where nodes are patches, and weighted edges connect nodes in neighboring frames. Let $\mathbf{I}$ be a set of frames of a video and $\mathbf{q}_t$ be the set of $N$ nodes extracted from frame $\mathbf{I}_t$, e.g. by sampling overlapping patches in a grid. An encoder $\phi$ maps nodes to $l_2$-normalized $d$-dimensional vectors, which we use to compute a pairwise similarity function $d_\phi(q_1, q_2) = \langle \phi(q_1), \phi(q_2) \rangle$ and an embedding matrix for $\mathbf{q}_t$ denoted $Q_t \in \mathbb{R}^{N \times d}$. We convert pairwise similarities into non-negative affinities by applying a softmax (with temperature $\tau$) over edges departing from each node. For timesteps $t$ and $t + 1$, the stochastic matrix of affinities is

$$A_t^{t+1}(i,j) = \texttt{softmax}(Q_t Q_{t+1}^\top)_{ij} = \frac{\exp(d_\phi(\mathbf{q}_t^i, \mathbf{q}_{t+1}^j)/\tau)}{\sum_{l=1}^N \exp(d_\phi(\mathbf{q}_t^i, \mathbf{q}_{t+1}^l)/\tau)}, \qquad (1)$$

where the `softmax` is row-wise. Note that this describes only the *local* affinity between the patches of two video frames, $\mathbf{q}_t$ and $\mathbf{q}_{t+1}$. The affinity matrix for the entire graph, which relates all nodes in the video as a Markov chain, is block-sparse and composed of local affinity matrices.

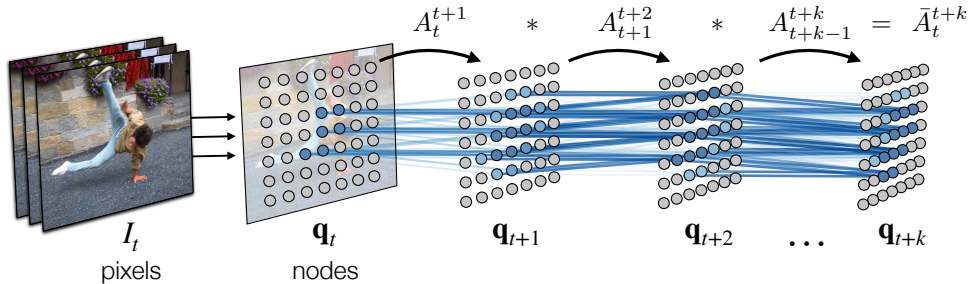

$I_t$
pixels

$\mathbf{q}_t$
nodes

$\mathbf{q}_{t+1}$

$\mathbf{q}_{t+2}$

$\dots$

$\mathbf{q}_{t+k}$

Figure 2: **Correspondence as a Random Walk**. We build a space-time graph by extracting nodes from each frame and allowing directed edges between nodes in neighboring frames. The transition probabilities of a random walk along this graph are determined by pairwise similarity in a learned representation.

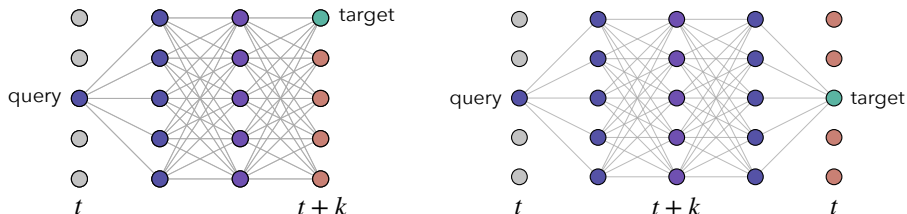

query

target

$t$

$t+k$

query

target

$t$

$t+k$

$t$

Figure 3: **Learning to Walk on Video.** (a) Specifying a target multiple steps in the future provides implicit supervision for *latent* correspondences along each path *(left)*. (b) We can construct targets for free by choosing palindromes as sequences for learning *(right)*.

Given the spatio-temporal connectivity of the graph, a step of a random walker on this graph can be viewed as performing tracking by *contrasting* similarity of neighboring nodes (using encoder $\phi$). Let $X_t$ be the state of the walker at time $t$, with transition probabilities $A_t^{t+1}(i,j) = P(X_{t+1} = j | X_t = i)$, where $P(X_t = i)$ is the probability of being at node $i$ at time $t$. With this view, we can formulate long-range correspondence as walking multiple steps along the graph (Figure 2):

$$\bar{A}_t^{t+k} = \prod_{i=0}^{k-1} A_{t+i}^{t+i+1} = P(X_{t+k} | X_t). \qquad (2)$$

**Guiding the walk.**   Our aim is to train the embedding to encourage the random walker to follow paths of corresponding patches as it steps through time. While ultimately we will train without labels, for motivation suppose that we did have ground-truth correspondence between nodes in two frames of a video, $t$ and $t+k$ (Figure 3a). We can use these labels to fit the embedding by maximizing the likelihood that a walker beginning at a *query* node at $t$ ends at the *target* node at time $t+k$:

$$\mathcal{L}_{sup} = \mathcal{L}_{CE}(\bar{A}_t^{t+k}, Y_t^{t+k}) = -\sum_{i=1}^{N} \log P(X_{t+k} = Y_t^{t+k}(i) | X_t = i), \qquad (3)$$

where $\mathcal{L}_{CE}$ is cross entropy loss and $Y_t^{t+k}$ are correspondence labels for matching time $t$ to $t+k$. Given the way transition probabilities are computed, the walk can be viewed as a chain of contrastive learning problems. Providing supervision at *every* step amounts to maximizing similarity between query and target nodes adjacent in time, while minimizing similarity to all other neighbors.

The more interesting case is supervision of longer-range correspondence, i.e. $k > 1$. In this case, the labels of $t$ and $t+k$ provide *implicit* supervision for intermediate frames $t+1, ..., t+k-1$, assuming that latent correspondences exist to link $t$ and $t+k$. Recall that in computing $P(X_{t+k} | X_t)$, we marginalize over all intermediate paths that link nodes in $t$ and $t+k$. By minimizing $\mathcal{L}_{sup}$, we shift affinity to paths that link the query and target. In easier cases (e.g. smooth videos), the paths that the walker takes from each node will not overlap, and these paths will simply be reinforced. In more ambiguous cases – e.g. deformation, multi-modality, or one-to-many matches – transition probability may be split across latent correspondences, such that we consider distribution over paths with higher entropy. The embedding should capture similarity between nodes in a manner that hedges probability over paths to overcome ambiguity, while avoiding transitions to nodes that lead the walker astray.

## 2.1 Self-Supervision

How to obtain query-target pairs that are known to correspond, without human supervision? We can consider training on graphs in which correspondence between the first and last frames are known, by construction. One such class of sequences are *palindromes*, i.e. sequences that are identical when reversed, for which targets are known since the first and last frames are identical. Given a sequence of frames $(I_t, ..., I_{t+k})$, we form training examples by simply concatenating the sequence with a temporally reversed version of itself: $(I_t, ...I_{t+k}, ...I_t)$. Treating each query node's position as its own target (Figure 3b), we obtain the following cycle-consistency objective:

$$\mathcal{L}_{cyc}^k = \mathcal{L}_{CE}(\bar{A}_t^{t+k}\bar{A}_{t+k}^t, I) = -\sum_{i=1}^{N} \log P(X_{t+2k} = i | X_t = i) \tag{4}$$

By leveraging structure in the graph, we can generate supervision for chains of contrastive learning problems that can be made arbitrarily long. As the model computes a soft attention distribution at every time step, we can backpropagate error across – and thus learn from – the many alternate paths of similarity that link query and target nodes.

**Contrastive learning with latent views.** To better understand the model, we can interpret it as contrastive learning with latent views. The popular InfoNCE formulation [74] draws the representation of two views of the same example closer by minimizing the loss $\mathcal{L}_{CE}(U_1^2, I)$, where $U_1^2 \in \mathbb{R}^{n \times n}$ is the normalized affinity matrix between the vectors of the first and second views of $n$ examples, as in Equation 1. Suppose, however, that we do not know *which* views should be matched with one another, merely that there should be a soft one-to-one alignment between them. We can formulate this as contrastive learning guided by a 'one-hop' cycle-consistency constraint, composing $U_1^2$ with the "transposed" stochastic similarity matrix $U_2^1$, to produce the loss $\mathcal{L}_{CE}(U_1^2 U_2^1, I)$, akin to Equation 4.

This task becomes more challenging with multiple hops, as avoiding spurious features that lead to undesirable diffusion of similarity across the graph becomes more important. While there are other ways of learning to align sets of features – e.g. by assuming soft bijection [18, 30, 114, 85] – it is unclear how they should extend to the multi-hop setting, where such heuristics may not always be desirable at each intermediate step. The proposed objective avoids the need to explicitly infer intermediate latent views, instead imposing a sequence-level constraint based on long-range correspondence known by construction.

## 2.2 Edge Dropout

One might further consider correspondence on the level of broader *segments*, where points within a segment have strong affinity to all other points in the segment. This inspires a trivial extension of the method – randomly dropping edges from the graph, thereby forcing the walker to consider alternative paths. We apply dropout [91] (with rate $\delta$) to the transition matrix $A$ to obtain $\tilde{A} = \mathtt{dropout}(A, \delta)$, and then renormalize. The resulting transition matrix $B$ and noisy cycle loss are:

$$B_{ij} = \frac{\tilde{A}_{ij}}{\sum_l \tilde{A}_{il}} \qquad \mathcal{L}_{c\tilde{y}c}^k = \mathcal{L}_{CE}(B_{t+k}^t B_t^{t+k}, I).$$

Edge dropout affects the task by randomly obstructing paths, thus encouraging hedging of mass to paths correlated with the ideal path – i.e. paths of *common fate* [111] – similar to the effect in spectral-based segmentation [89, 67]. In practice, we apply edge dropout before normalizing affinities, by setting values to a negative constant. We will see in Section 3.2 that edge dropout improves object-centric correspondence.

## 2.3 Implementation

We now describe how we construct the graph and parameterize the node embedding $\phi$. Algorithm 1 provides complete pseudocode for the method.

**Pixels to Nodes.** At training time, we follow [43], where patches of size $64 \times 64$ are sampled on a $7 \times 7$ grid from a $256 \times 256$ image (i.e. 49 nodes per frame). Patches are spatially jittered to prevent matching based on borders. At test time, we found that we could reuse the convolutional feature map between patches instead of processing the patches independently [59], making the features computable with only a single feed-forward pass of our network.[1]

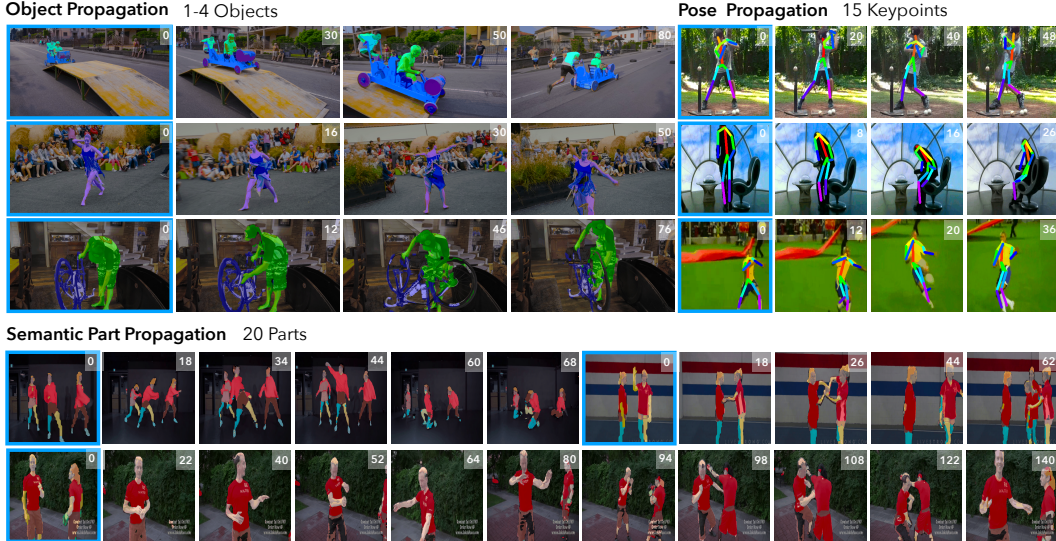

**Object Propagation** 1-4 Objects      **Pose Propagation** 15 Keypoints

**Semantic Part Propagation** 20 Parts

Figure 4: Qualitative results for label propagation under our model for object, pose, and semantic part propagation tasks. The first frame is indicate with a blue outline. **Please see our webpage for video results**, as well as a qualitative comparison with other methods.

**Encoder $\phi$.** We create an embedding for each image patch using a convolutional network, namely ResNet-18 [42]. We apply a linear projection and $l_2$ normalization after average pooling, obtaining a 128-dimensional vector. We reduce the stride of last two residual blocks (`res3` and `res4`) to be 1. Please see Appendix G for details.

**Shorter paths.** During training, we consider paths of multiple lengths. For a sequence of length $T$, we optimize all *sub*-cycles: $\mathcal{L}_{train} = \sum_{i=1}^{T} \mathcal{L}_{cyc}^i$. This loss encourages the sequence of nodes visited in the walk to be a palindrome, i.e. on a walk of length $N$, the node visited at step $t$ should be the same node as $N - t$. It induces a curriculum, as short walks are easier to learn than long ones. This can be computed efficiently, since the losses share affinity matrices.

**Training.** We train $\phi$ using the (unlabeled) videos from Kinetics400 [14], with Algorithm 1.

**Algorithm 1** Pseudocode in a PyTorch-like style.

```
for x in loader: # x: batch with B sequences
  # Split image into patches
  # B x C x T x H x W -> B x C x T x N x h x w
  x = unfold(x, (patch_size, patch_size))
  x = spatial_jitter(x)
  # Embed patches (B x C x T x N)
  v = l2_norm(resnet(x))

  # Transitions from t to t+1 (B x T-1 x N x N)
  A = einsum("bcti,bctj->btij",
             v[:,:,:-1], v[:,:,1:]) / temperature

  # Transition energies for palindrome graph
  AA = cat((A, A[:,:,:-1].transpose(-1,-2), 1)
  AA[rand(AA) < drop_rate] = -1e10 # Edge dropout
  At = eye(P)                      # Init. position

  # Compute walks
  for t in range(2*T-2):
    At = bmm(softmax(AA[:,t]), dim=-1), At)

  # Target is the original node
  loss = At[[range(P)]*B]].log()
```

bmm: batch matrix multiplication; `eye`: identity matrix; `cat`: concatenation.; `rand`: random tensor drawn from $(0, 1)$.

We used the Adam optimizer [49] for two million updates with a learning rate of $1 \times 10^{-4}$. We use a temperature of $\tau = 0.07$ in Equation 1, following [113] and resize frames to $256 \times 256$ (before extracting nodes, as above). Except when indicated otherwise, we report results with edge dropout rate 0.1 and a videos of length 10. Please find more details in Appendix E.

## 3 Experiments

We evaluate the learned representation on video label propagation tasks involving objects, keypoints, and semantic parts, by using it as a similarity metric. We also study the effects of edge dropout, training sequence length, and self-supervised adaptation at test-time. In addition to comparison with the state-of-the-art, we consider a baseline of label propagation with strong pre-trained features. Please find additional details, comparisons, ablations, and qualitative results in the Appendices.

### 3.1 Transferring the Learned Representation

We transfer the trained representation to label propagation tasks involving objects, semantic parts, and human pose. To isolate the effect of the representation, we use a simple inference algorithm based on $k$-nearest neighbors. Qualitative results are shown in Figure 4.

| Method | Resolution | Train Data | $\mathcal{J}\&\mathcal{F}_{\mathrm{m}}$ | $\mathcal{J}_{\mathrm{m}}$ | $\mathcal{J}_{\mathrm{r}}$ | $\mathcal{F}_{\mathrm{m}}$ | $\mathcal{F}_{\mathrm{r}}$ |
|---|---|---|---|---|---|---|---|
| ImageNet [42] | 1× | ImageNet | 62.9 | 60.6 | 69.9 | 65.2 | 73.8 |
| MoCo [40] | 1× | ImageNet | 60.8 | 58.6 | 68.7 | 63.1 | 72.7 |
| VINCE [31] | 1× | Kinetics | 60.4 | 57.9 | 66.2 | 62.8 | 71.5 |
| CorrFlow⋆ [56] | 2× | OxUvA | 50.3 | 48.4 | 53.2 | 52.2 | 56.0 |
| MAST⋆ [55] | 2× | OxUvA | 63.7 | 61.2 | 73.2 | 66.3 | 78.3 |
| MAST⋆ [55] | 2× | YT-VOS | 65.5 | 63.3 | 73.2 | 67.6 | 77.7 |
| TimeCycle [109] | 1× | VLOG | 48.7 | 46.4 | 50.0 | 50.0 | 48.0 |
| UVC+track⋆ [57] | 1× | Kinetics | 59.5 | 57.7 | 68.3 | 61.3 | 69.8 |
| UVC [57] | 1× | Kinetics | 60.9 | 59.3 | 68.8 | 62.7 | 70.9 |
| **Ours** w/ dropout | 1× | Kinetics | **67.6** | **64.8** | **76.1** | **70.2** | **82.1** |
| w/ dropout & adaptation | 1× | Kinetics | 68.3 | 65.5 | 78.6 | 71.0 | 82.9 |

Table 1: **Video object segmentation results on DAVIS 2017 val set** Comparison of our method (2 variants), with previous self-supervised approaches and strong pretrained feature baselines. *Resolution* indicates if the approach uses a high-resolution (2x) feature map. *Train Data* indicates which dataset was used for pre-training. $\mathcal{F}$ is a boundary alignment metric, while $\mathcal{J}$ measures region similarity as IOU between masks. ⋆ indicates that our label propagation algorithm is not used.

**Label propagation.** All evaluation tasks considered are cast as video label propagation, where the task is to predict labels for each pixel in *target* frames of a video given only ground-truth for the first frame (i.e. the *source*). We use the representation as a similarity function for prediction by $k$-nearest neighbors, which is natural under our model and follows prior work for fair comparison [109, 57].

Say we are given source nodes $\mathbf{q}_s$ with labels $L_s \in \mathbb{R}^{N \times C}$, and target nodes $\mathbf{q}_t$. Let $K_t^s$ be the matrix of transitions between $\mathbf{q}_t$ and $\mathbf{q}_s$ (Equation 1), with the special property that only the top$-k$ transitions are considered per target node. Labels $L_t$ are propagated as $L_t = K_t^s L_s$, where each row corresponds to the soft distribution over labels for a node, predicted by $k$-nearest neighbor in $d_\phi$.

To provide temporal context, as done in prior work [109, 56, 57], we use a queue of the last $m$ frames. We also restrict the set of source nodes considered to a spatial neighborhood of the query node for efficiency (i.e. *local* attention). The source set includes nodes of the first labeled frame, as well as the nodes in previous $m$ frames, whose predicted labels are used for auto-regressive propagation. The softmax computed for $K_t^s$ is applied over all source nodes. See Appendix F for further discussion and hyper-parameters.

**Baselines.** All baselines use ResNet-18 [42] as the backbone, modified to increase spatial resolution of the feature map by reducing the stride of the last two residual blocks to be 1. For consistency across methods, we use the output of the penultimate residual block as node embeddings at test-time.

Pre-trained visual features: We evaluate pretrained features from strong image- and video-based representation learning methods. For a strongly supervised approach, we consider a model trained for classification on **ImageNet** [20]. We also consider a strong self-supervised method, **MoCo** [40]. Finally, we compare with a video-based contrastive learning method, **VINCE** [31], which extends MoCo to videos (Kinetics) with views from data augmentation *and* neighbors in time.

Task-specific approaches: **Wang et al.** [109] uses cycle-consistency to train a spatial transformer network as a deterministic patch tracker. We also consider methods based on the **Colorization** approach of Vondrick et al. [104], including high-resolution methods: **CorrFlow** [56] and **MAST** [55]. CorrFlow combines cycle consistency with colorization. MAST uses a deterministic region localizer and memory bank for high-resolution colorization, and performs multi-stage training on [98]. Notably, both [56, 55] use feature maps that are significantly higher resolution than other approaches (2×) by removing max pooling from the network. Finally, **UVC** [57] jointly optimizes losses for colorization, grouping, pixel-wise cycle-consistency, and patch tracking with a deterministic patch localizer.

### 3.1.1 Video Object Segmentation

We evaluate our model on DAVIS 2017 [83], a popular benchmark for video object segmentation, for the task of semi-supervised multi-object (i.e. 2-4) segmentation. Following common practice, we evaluate on 480p resolution images. We apply our label propagation algorithm for all comparisons, except CorrFlow and MAST [56, 55], which require 4× more GPU memory. We report mean (m) and recall (r) of standard boundary alignment ($\mathcal{F}$) and region similarity ($\mathcal{J}$) metrics, detailed in [78].

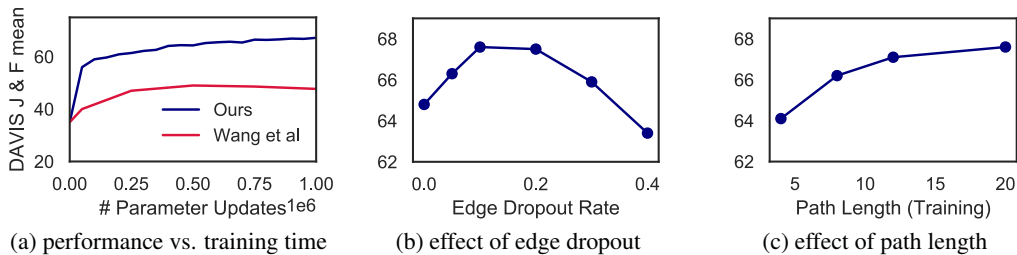

(a) performance vs. training time  (b) effect of edge dropout  (c) effect of path length

Figure 5: **Variations of the Model.** (a) Downstream task performance as a function of training time. (b) Moderate edge dropout improves object-level correspondences. (c) Training on longer paths is beneficial. All evaluations are on the DAVIS segmentation task.

As shown in Table 1, our approach outperforms other self-supervised methods, without relying on machinery such as localization modules or multi-stage training. We also outperform [55] despite being more simple at train and test time, and using a lower-resolution feature map. We found that when combined with a properly tuned label propagation algorithm, the more generic pretrained feature baselines fare better than more specialized temporal correspondence approaches. Our approach outperformed approaches such as MoCo [40] and VINCE [31], suggesting that it may not always be optimal to choose views for contrastive learning by random crop data augmentation of frames. Finally, our model compares favorably to many supervised approaches with architectures designed for dense tracking [78, 12, 106] (see Appendix B).

### 3.1.2 Pose Tracking

We consider pose tracking on the JHMDB benchmark, which involves tracking 15 keypoints. We follow the evaluation protocol of [57], using $320 \times 320$px images. As seen in Table 2, our model outperforms existing self-supervised approaches, including video colorization models that directly optimize for fine-grained matching with pixel-level objectives [57]. We attribute this success to the fact that our model sees sufficiently hard negative samples drawn from the same image at training time to learn features that discriminate beyond color. Note that our inference procedure is naive in that we propagate keypoints independently, without leveraging relational structure between them.

### 3.1.3 Video Part Segmentation

We consider the semantic part segmentation task of the Video Instance Parsing (VIP) benchmark [120], which involves propagating labels of 20 parts — such as arm, leg, hair, shirt, hand — requiring more precise correspondence than DAVIS. The sequences are longer and sampled at a lower frame rate. We follow the evaluation protocol of [57], using $560 \times 560$px images and $m = 1$. The model outperforms existing self-supervised methods, and when using more temporal context (i.e. $m = 4$), outperforms the baseline supervised approach of [120].

|  | Parts | Pose | |
| --- | --- | --- | --- |
| Method | mIoU | PCK@0.1 | PCK@0.2 |
| TimeCycle [109] | 28.9 | 57.3 | 78.1 |
| UVC [57] | 34.1 | 58.6 | 79.6 |
| Ours | 36.0 | 59.0 | 83.2 |
| Ours + context | **38.6** | **59.3** | **84.9** |
| ImageNet [42] | 31.9 | 53.8 | 74.6 |
| ATEN [120] | 37.9 | – | – |
| Yang et al. [115] | – | **68.7** | **92.1** |

Table 2: **Part and Pose Propagation tasks**, with the VIP and JHMDB benchmarks, respectively. For comparison, we show supervised methods below.

### 3.2 Variations of the Model

**Edge dropout.** We test the hypothesis (Figure 5b) that edge dropout should improve performance on the object segmentation task, by training our model with different edge dropout rates: {0, 0.05, 0.1, 0.2, 0.3, 0.4}. Moderate edge dropout yields a significant improvement on the DAVIS benchmark. Edge dropout simulates partial occlusion, forcing the network to consider reliable context.

**Path length.** We also asked how important it is for the model to see longer sequences during training, by using clips of length 2, 4, 6, or 10 (resulting in paths of length 4, 8, 12, or 20). Longer sequences yield harder tasks due to compounding error. We find that longer training sequences accelerated convergence as well as improved performance on the DAVIS task (Figure 5c). This is in contrast to prior work [109]; we attribute this success to considering multiple paths at training time via soft-attention, which allows for learning from longer sequences, despite ambiguity.

**Improvement with training** We found that the model's downstream performance on DAVIS improves as more data is seen during self-supervised training (Figure 5a). Compared to Wang et al [109], there is less indication of saturation of performance on the downstream task.

### 3.3 Self-supervised Adaptation at Test-time

A key benefit of not relying on labeled data is that training need not be limited to the training phase, but can continue during deployment [3, 70, 93]. Our approach is especially suited for such adaptation, given the non-parametric inference procedure. We ask whether the model can be improved for object correspondence by fine-tuning the representation *at test time* on a novel video. Given an input video, we can perform a small number of iterations of gradient descent on the self-supervised loss (Algorithm 1) *prior* to label propagation. We argue it is most natural to consider an online setting, where the video is ingested as a stream and fine-tuning is performed continuously on the sliding window of $k$ frames around the current frame. Note that only the raw, unlabeled video is used for this adaptation; we do not use the provided label mask. As seen in Table 1, test-time training improves object propagation. Interestingly, we see most improvement in the recall of the region similarity metric $\mathcal{J}_{recall}$ (which measures how often more than 50% of the object is segmented). More experiment details can be found in Appendix E.

## 4    Related Work

**Temporal Correspondence.** Many early methods represented video as a spatio-temporal $XYT$ volume, where patterns, such as lines or statistics of spatio-temporal gradients, were computed for tasks like gait tracking [71] and action recognition [117]. Because the camera was usually static, this provided an implicit temporal correspondence via $(x, y)$ coordinates. For more complex videos, optical flow [62] was used to obtain short-range explicit correspondences between patches of neighboring frames. However, optical flow proved too noisy to provide long-range composite correspondences across many frames. Object tracking was meant to offer robust long-range correspondences for a given tracked object. But after many years of effort (see [27] for overview), that goal was largely abandoned as too difficult, giving rise to "tracking as repeated detection" paradigm [84], where trained object detectors are applied to each frame independently. In the case of multiple objects, the process of "data association" resolves detections into coherent object tracks. Data association is often cast as an optimization problem for finding paths through video that fulfill certain constraints, e.g. appearance, position overlap, etc. Approaches include dynamic programming, particle filtering, various graph-based combinatorial optimization, and more recently, graph neural networks [118, 86, 8, 82, 15, 116, 48, 47, 53, 11, 13, 51]. Our work can be seen as contrastive data association via soft-attention, as a means for learning representations directly from pixels.

**Graph Neural Networks and Attention.** Representing inputs as graphs has led to unified deep learning architectures. Graph neural networks – versatile and effective across domains [100, 37, 52, 101, 107, 6, 13] – can be seen as learned message passing algorithms that iteratively update node representations, where propagation of information is dynamic, contingent on local and global relations, and often implemented as soft-attention. Iterative routing of information encodes structure of the graph for downstream tasks. Our work uses cross-attention between nodes of adjacent frames to learn to propagate node identity through a graph, where the task – in essence, instance discrimination across space and time – is designed to induce representation learning.

**Graph Partitioning.** Graphs have been widely used in image and video segmentation as a data structure. Given a video, a graph is formed by connecting pixels in spatio-temporal neighborhoods, followed by spectral clustering [88, 89, 28] or MRF/GraphCuts [10]. Most relevant is the work of Meila and Shi [67], which poses Normalized Cuts as a Markov random walk, describing an algorithm for learning an affinity function for segmentation by fitting the transition probabilities to be uniform within segments and zero otherwise. More recently, there has been renewed interest in the problem of unsupervised grouping [34, 50, 33, 25, 51]. Many of these approaches can be viewed as end-to-end neural architectures for graph partitioning, where entities are partitions of images or video inferred by learned clustering algorithms or latent variable models implemented with neural networks. While these approaches explicitly group without supervision, they have mainly considered simpler data. Our work similarly aims to model groups in dynamic scenes, but does so implicitly so as to scale to real, large-scale video data. Incorporating more explicit entity estimation is an exciting direction.

**Graph Representation Learning.** Graph representation learning approaches solve for distributed representations of nodes and vertices given connectivity in the graph [38]. Most relevant are similarity learning approaches, which define neighborhoods of positives with fixed (i.e. $k$-hop neighborhood) or stochastic (i.e. random walk) heuristics [79, 35, 94, 37], while sampling negatives at random. Many of these approaches can thus be viewed as fitting shallow graph neural networks with tasks reminiscent of Mikolov et al. [68]. Backstrom et al. [5] learns to predict links by supervising a

random walk on social network data. While the above consider learning representations given a single graph, others have explored learning node embeddings given multiple graphs. A key challenge is inferring correspondence between graphs, which has been approached in prior work [114, 85] with efficient optimal transport algorithms [90, 18, 80]. We use graph matching as a means for representation learning, using cycle-consistency to supervise a chain of matches, without inferring correspondence between intermediate pairs of graphs. In a similar vein, cycle-consistency has also been shown to be a useful constraint for solving large-scale optimal transport problems [61].

**Self-supervised Visual Representation Learning.** Most work in self-supervised representation learning can be interpreted as data imputation: given an example, the task is to predict a part — or *view* — of its data given another view [7, 19, 17]. Earlier work leveraged unlabeled visual datasets by constructing *pretext* prediction tasks [21, 72, 119]. For video, temporal information makes for natural pretext tasks, including future prediction [32, 92, 66, 60, 64], arrow of time [69, 110], motion estimation [1, 46, 97, 58] or audio [76, 2, 75, 54]. The use of off-the-shelf tracking to provide supervisory signal for learning visual similarity has also been explored [108, 77]. Recent progress in self-supervised learning has focused on improving techniques for large-scale deep similarity learning, e.g. by combining the cross-entropy objective with negative sampling [36, 68]. Sets of corresponding views are constructed by composing combinations of augmentations of the same instance [22, 9, 113], with domain knowledge being crucial for picking the right data augmentations. Strong image-level visual representations can be learned by heuristically choosing views that are close in space [99, 44, 4, 40, 16], in time [87, 81, 39, 96, 31] or both [45, 95], even when relying on noisy negative samples. However, forcing random crops to be similar is not always desirable because they may not be in correspondence. In contrast, we implicitly determine which views to bring closer – a sort of automatic view selection.

**Self-supervised Correspondence and Cycle-consistency.** Our approach builds on recent work that uses cycle-consistency [121, 23] in time as supervisory signal for learning visual representations from video [109, 105]. The key idea in [109, 105] is to use self-supervised tracking as a pretext task: given a patch, first track forward in time, then backward, with the aim of ending up where it started, forming a cycle. These methods rely on trackers with hard attention, which limits them to sampling, and learning from, one path at a time. In contrast, our approach computes soft-attention at every time step, considering many paths to obtain a dense learning signal and overcome ambiguity. Li et al. [57] combines patch tracking with other losses including color label propagation [104], grouping, and cycle-consistency via an orthogonality constraint [29], considering pairs of frames at a time. Lai et al. [56, 55] refine architectural and training design decisions that yield impressive results on video object segmentation and tracking tasks. While colorization is a useful cue, the underlying assumption that corresponding pixels have the same color is often violated, e.g. due to lighting or deformation. In contrast, our loss is discriminative and permits association between regions that may have significant differences in their appearance.

## 5 Discussion

While data augmentation can be tuned to induce representation learning tasks involving invariance to color and local context, changes in other important factors of variation – such as physical transformations – are much harder to simulate. We presented a self-supervised approach for learning representations for space-time correspondence from unlabeled video data, based on learning to walk on a space-time graph. Under our formulation, a simple path-level constraint provides implicit supervision for a chain of contrastive learning problems. Our learning objective aims to leverage the natural data augmentation of dynamic scenes, *i.e.* how objects change and interact over time, and can be combined with other learning objectives. Moreover, it builds on a connection between self-supervised representation learning and unsupervised grouping [67]. As such, we hope this work is a step toward learning to discover and describe the structure and dynamics of natural scenes from large-scale unlabeled video.

## 6 Broader Impact

Research presented in the paper has a potential to positively contribute to a number of practical applications where establishing temporal correspondence in video is critical, among them pedestrian safely in automotive settings, patient monitoring in hospitals and elderly care homes, video-based animal monitoring and 3D reconstruction, etc. However, there is also a potential for the technology to be used for nefarious purposes, mainly in the area of unauthorized surveillance, especially by

autocratic regimes. As partial mitigation, we commit to not entering into any contracts involving this technology with any government or quasi-governmental agencies of countries with an *EIU Democracy Index* [24] score of $4.0$ or below ("authoritarian regimes"), or authorizing them to use our software.

**Acknowledgments.** We thank Amir Zamir, Ashish Kumar, Yu Sun, Tim Brooks, Bill Peebles, Dave Epstein, Armand Joulin, and Jitendra Malik for helpful feedback and support. We are also grateful to the wonderful members of VGG for hosting us during a dreamy semester at Oxford. This work would not have been possible without the hospitality of Port Meadow and the swimming pool on Iffley Road. Research was supported, in part, by NSF grant IIS-1633310, the DARPA MCS program, and NSF IIS-1522904. We are grateful for compute resources donated by NVIDIA. AJ is supported by the PD Soros Fellowship.

## Footnotes

[1]Using a single convolutional feature map for training was susceptible to shortcut solutions; see Appendix C.

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
