[Supplementary Material]

## A  Label Noise: Effect of Identical Patches

Here, we show that false negatives that are identical to the positive – for example, patches of the sky – do not change the sign of gradient associated with the positive. Let $q$ be the query, $u$ be the positive, $V$ be the set of negatives. W.l.o.g, let the softmax temperature $\tau = 1$. The loss and corresponding gradient can be expressed as follows, where $Z$ is the partition function:

$$L(q, u, V) = u^\top q - \log[\exp u^\top q + \sum_{v \in V} \exp v^\top q] = u^\top q - \log Z$$

$$\nabla_q L(q, u, V) = u - \frac{\exp u^\top q}{Z} u - \sum_{v \in V} \frac{\exp v^\top q}{Z} v = (1 - \frac{\exp u^\top q}{Z}) u - \sum_{v \in V} \frac{\exp v^\top q}{Z} v$$

Let $V^-$ be the set of false negatives, such that $V^- \subseteq V$ and $V^+ = V \setminus V^-$. Consider the worst case, whereby $v_- = u, \forall v_- \in V^-$, so that false negatives are exactly identical to the positive:

$$\nabla_q L(q, u, V) = (1 - \frac{\exp u^\top q}{Z}) u - \sum_{v_- \in V^-} \frac{\exp v_-^\top q}{Z} v_- - \sum_{v_+ \in V^+} \frac{\exp v_+^\top q}{Z} v_+$$

$$= \underbrace{\left(1 - \frac{(1 + |V^-|) \exp u^\top q}{Z}\right)}_{\lambda_u} u - \sum_{v_+ \in V^+} \frac{\exp v_+^\top q}{Z} v_+$$

It is easy to see that the contribution of the negatives that are identical to the positive do not reverse the sign of the positive gradient, i.e. $\lambda_u \geq 0$, so that in the worse case the gradient vanishes:

$$\lambda_u = 1 - \frac{(1 + |V^-|) \exp u^\top q}{Z}$$

$$= 1 - \frac{(1 + |V^-|) \exp u^\top q}{(1 + |V^-|) \exp u^\top q + \sum_{v_+ \in V^+} \exp v_+^\top q}$$

$$\geq 0$$

## B  Comparison to Supervised Methods on DAVIS-VOS

The proposed method outperforms many supervised methods for video object segmentation, despite relying on a simple label propagation algorithm, not being trained for object segmentation, and not training on the DAVIS dataset. We also show comparisons to pretrained feature baselines with larger networks.

| Method | Backbone | Train Data (#frames) | $\mathcal{J}\&\mathcal{F}_m$ | $\mathcal{J}_m$ | $\mathcal{J}_r$ | $\mathcal{F}_m$ | $\mathcal{F}_r$ |
|---|---|---|---|---|---|---|---|
| OSMN [115] | VGG-16 | I/C/D (1.2M + 227k) | 54.8 | 52.5 | 60.9 | 57.1 | 66.1 |
| SiamMask [106] | ResNet-50 | I/V/C/Y (1.2M + 2.7M) | 56.4 | 54.3 | 62.8 | 58.5 | 67.5 |
| OSVOS [12] | VGG-16 | I/D (1.2M + 10k) | 60.3 | 56.6 | 63.8 | 63.9 | 73.8 |
| OnAVOS [103] | ResNet-38 | I/C/P/D (1.2M + 517k) | 65.4 | 61.6 | 67.4 | 69.1 | 75.4 |
| OSVOS-S [65] | VGG-16 | I/P/D (1.2M + 17k) | 68.0 | 64.7 | 74.2 | 71.3 | 80.7 |
| FEELVOS [102] | Xception-65 | I/C/D/Y (1.2M + 663k) | 71.5 | 69.1 | 79.1 | 74.0 | 83.8 |
| PReMVOS [63] | ResNet-101 | I/C/D/P/M (1.2M + 527k) | 77.8 | 73.9 | 83.1 | 81.8 | 88.9 |
| STM [73] | ResNet-50 | I/D/Y (1.2M + 164k) | 81.8 | 79.2 | - | 84.3 | - |
| ImageNet [41] | ResNet-50 | I (1.2M) | 66.0 | 63.7 | 74.0 | 68.4 | 79.2 |
| MoCo [40] | ResNet-50 | I (1.2M) | 65.4 | 63.2 | 73.0 | 67.6 | 78.7 |
| **Ours** | ResNet-18 | K (20M unlabeled) | 67.6 | 64.8 | 76.1 | 70.2 | 82.1 |

Table 3: **Video object segmentation results on DAVIS 2017 val set**. We show results of state-of-the-art **supervised** approaches in comparison to our unsupervised one (see main paper for comparison with unsupervised methods). Key for *Train Data* column: I=ImageNet, K=Kinetics, V = ImageNet-VID, C=COCO, D=DAVIS, M=Mapillary, P=PASCAL-VOC Y=YouTube-VOS. $\mathcal{F}$ is a boundary alignment metric, while $\mathcal{J}$ measures region similarity as IOU between masks.

# C  Using a Single Feature Map for Training

We follow the simplest approach for extracting nodes from an image without supervision, which is to simply sample patches in a convolutional manner. The most efficient way of doing this would be to only encode the image once, and pool the features to obtain region-level features [59].

We began with that idea and found that the network could cheat to solve this dense correspondence task even across long sequences, by learning a shortcut. It is well-known that convolutional networks can learn to rely on boundary artifacts [59] to encode position information, which is useful for the dense correspondence task. To control for this, we considered: 1) removing padding altogether; 2) reducing the receptive field of the network to the extent that entries in the center crop of the spatial feature map do not see the boundary; we then cropped the feature map to only see this region; 3) randomly blurring frames in each video to combat space-time compression artifacts; and 4) using *random* videos made of noise. Surprisingly, the network was able to learn a shortcut in each case. In the case of random videos, the shortcut solution was not nearly as successful, but we still found it surprising that the self-supervised loss of Equation 2.2 could be optimized at all.

# D  Frame-rate Ablation

**Effect of frame-rate at training time**   We ablate the effect of frame-rate (i.e. frames per second) used to generate sequences for training, on downstream object segmentation performance. The case of infinite frame-rate corresponds to the setting where the *same* image is used in each time step; this experiment is meant to disentangle the effect of data augmentation (spatial jittering of patches) from the natural "data augmentation" observed in video. We observe that spatio-temporal transformations is beneficial for learning of representations that transfer better for object segmentation.

| Frame rate | $\mathcal{J}\&\mathcal{F}_\mathrm{m}$ |
|---|---|
| 2 | 65.9 |
| 4 | 67.5 |
| 8 | 67.6 |
| 30 | 62.3 |
| $\infty$ | 57.5 |

# E  Hyper-parameters

We list the key hyper-parameters and ranges considered at training time.  Due to computational constraints, we did not tune the patch extraction strategy, nor several other hyper-parameters. The hyper-parameters varied, namely edge dropout and video length, were ablated in Section 3 (shown in bold). Note that the effective training path length is twice that of the video sequence length.

| *Train* Hyper-parameters | Values |
|---|---|
| Learning rate | 0.0001 |
| Temperature $\tau$ | 0.07 |
| Dimensionality $d$ of embedding | 128 |
| Frame size | 256 |
| Video length | **2, 4, 6, 10** |
| Edge dropout | **0, 0.05, 0.1, 0.2, 0.3** |
| Frame rate | **2, 4, 8, 30** |
| Patch Size | 64 |
| Patch Stride | 32 |
| Spatial Jittering (crop range) | (0.7, 0.9) |

We tuned test hyper-parameters with the ImageNet baseline. In general, we found performance to increase given more context.  Here, we show hyper-parameters used in reported experiments; we largely follow prior work, but for the case of DAVIS, we used 20 frames of context.

| *Test* Hyper-parameters | Values |
|---|---|
| Temperature $\tau$ | 0.07 |
| Number of neighbors $k$ | **10**, 20 |
| Number of context frames $m$ | Objects: 20 |
| | Pose: 7 |
| | Parts: 4 |
| Spatial radius of source nodes | **12**, 20 |

# F   Label Propagation

We found that the performance of baselines can be improved by carefully implementing label propagation by $k$-nearest neighbors. When compared to baseline results reported in [57] and [55], the differences are:

1. Restricting the set of source nodes (context) considered for each target node, on the basis of spatial locality, i.e. *local* attention. This leads to a gain of $+4\%$ J&F for the ImageNet baseline.

    Many of the task-specific approaches for temporal correspondence incorporate restricted attention, and we found this rudimentary form to be effective and reasonable.

2. Computing attention over all source nodes at once and selecting the top-$k$, instead of independently selecting the top-$k$ from each frame. This leads to a gain of $+3\%$ J&F for the ImageNet baseline.

    This is more natural than computing nearest neighbors in each frame individually, and can be done efficiently if combined with local attention. Note that the softmax over context can be performed after nearest neighbors retrieval, for further efficiency.

## F.1   Effect of Label Propagation Hyper-parameters

We study the effect of hyper-parameters of the label propagation algorithm, when applied with strong baselines and our method. The key hyper-parameters are the length of context $m$, the number of neighbors $k$, and the search radius $r$. In the figures above, we see the benefit of adding context (see left, with $k = 10, r = 12$), effect of considering more neighbors (middle, with $r = 12$), and effect of radius (right, with $m = 20$).

# G   Encoder Architecture

We use the ResNet-18 network architecture [42], modified to increase the spatial resolution of the convolutional feature map. Specifically, we modify the stride of convolutions in the last two residual blocks to be 1. This increases the resolution by a factor of four, so that the downsampling factor is $1/8$. Please refer to Table 4 for a detailed description.

For evaluation, when applying our label propagation algorithm, we report results using the output of `res3` as node embeddings, for fair comparison to pretrained feature baselines ImageNet, MoCo, and VINCE, which were trained with stride 2 in `res3` and `res4`. We also found that `res3` features compared favorably to `res4` features.

| Layer | Output | Details |
|---|---|---|
| input | $H \times W$ | |
| conv1 | $H/2 \times W/2$ | $7 \times 7$, 64, stride 2 |
| maxpool | $H/4 \times W/4$ | stride 2 |
| res1 | $H/4 \times W/4$ | $\begin{bmatrix} 3 \times 3, 64 \\ 3 \times 3, 64 \end{bmatrix} \times 2$, stride 1 |
| res2 | $H/8 \times W/8$ | $\begin{bmatrix} 3 \times 3, 128 \\ 3 \times 3, 128 \end{bmatrix} \times 2$, stride 2 |
| res3 | $H/8 \times W/8$ | $\begin{bmatrix} 3 \times 3, 256 \\ 3 \times 3, 256 \end{bmatrix} \times 2$, stride ~~2~~ 1 |
| res4 | $H/8 \times W/8$ | $\begin{bmatrix} 3 \times 3, 512 \\ 3 \times 3, 512 \end{bmatrix} \times 2$, stride ~~2~~ 1 |

Table 4: Modified ResNet-18 Architecture. Our modifications are shown in blue.

# H   Test-time Training Details

We adopt the same hyper-parameters for optimization as in training: we use the Adam optimizer with learning rate 0.0001. Given an input video $I$, we fine-tune the model parameters by applying Algorithm 1 with input frames $\{I_{t-m}, ..., I_t, ..., I_{t+m}\}$, *prior* to propagating labels to $I_t$. For efficiency, we only finetune the model every 5 timesteps, applying Adam for 100 updates. In practice, we use $m = 10$, which we did not tune.

# I   Utility Functions used in Algorithm 1

**Algorithm 2** Utility functions.

```
// psize : size of patches to be extracted

import torch
import kornia.augmentation as K

# Turning images into list of patches
unfold = torch.nn.Unfold((psize, psize), stride=(psize//2, psize//2))

# l2 normalization
l2_norm = lambda x: torch.nn.functional.normalize(x, p=2, dim=1)

# Slightly cropping patches once extracted
spatial_jitter = K.RandomResizedCrop(size=(psize, psize), scale=(0.7, 0.9), ratio=(0.7, 1.3))
```