[Reviews · NeurIPS 2020]

Review 1

Summary and Contributions: This paper proposes a simple yet effective approach for learning space-time correspondence from videos. It learns to estimate the pairwise similarities between patches in adjacent frames via soft attention (Figure 2&3), then propagate the similarities first forward and then backward in time for the same number of steps. This naturally forms a cycle where the most probable end patch should the starting patch. This signal is used as "self supervision" via contrastive learning. The proposed method performs favorably when evaluated on tasks that require visual correspondence. ********** Post rebuttal: After reading the rebuttal and other reviews, I retain my original rating of "7: good submission" :)

Strengths: + Learning space-time correspondence is an important topic in computer vision which would be useful for many downstream recognition tasks. Self-supervised learning is highly necessary as manual annotation is costly and infeasible. + The idea of soft tracking on patches, and indirect supervision with only the final targets are elegant, and the proposed method is simple yet effective compared with the baseline methods, which I believe is a main strength of this paper, and make the idea potentially generalizable to other relevant tasks. + The authors provided rich visualizations and analysis of the results. The ablations on edge dropout rate and especially on path length are important to understand the behavior of the proposed algorithm.

Weaknesses: - Some important analysis are missing: for example, what are the typical failure modes of the proposed method? What would happen if an object moves out of camera, or there are textureless patches (e.g. sky, water) that are confusing to learn from? Could the visual encoder learn to implicitly encode relative spatial location in each frame and use that as a shortcut? If so how to avoid it? - I found the interpretation of "contrastive learning with latent views" in section 2 a bit confusing: on one hand, the "views" presented in this paper are not latent, but well-defined visual patches on different frames; on the other, it's unclear what properties the representations of the latent views need to have. For example, when the two views have the same representations, \mathcal{L}_{contrast} can be effectively minimized without capturing the semantics. It would be great if the authors could further elaborate on this interpretation. - Why choose 7x7 as the training patch size? What would be the effect on performance and speed by increasing or decreasing this patch size? In Figure 4, the propagated masks seem quite high-resolution. How was this achieved?

Correctness: The empirical methodology appears to be correct to me, I'm not sure on the latent view interpretation as outlined in section 2.

Clarity: Yes, the paper is very well written.

Relation to Prior Work: Mostly. It would be beneficial to the readers if relevant work on attention and graph neural networks are discussed.

Reproducibility: Yes

Additional Feedback:


Review 2

Summary and Contributions: This paper proposed a video-based self-supervised visual representation learning framework that casts a video as a spatio-temporal graph and contrastively predicts cycle-consistent paths of correspondence according to the mechanism of the random walk. In this graph, nodes are patches sampled in each frame and the edges are affinities between nodes of neighboring frames. Cycle-consistency unrolls the video as a palindrome, and it gives rich self-supervisory signals that each patch in the starting frame will have positive paths that terminate at the same patch at the ending frame of the palindrome, while leaves the paths terminate at other patches as the negative samples. The proposed method, in comparison to existing self-supervised methods that learn discriminative correspondences, considers all patchs and all paths simultaneously, making the learned representation more effective for a variety of correspondence tasks. The proposed method is simple but elegant, and scales effectively when trained with longer lengths of random walks.

Strengths: - This paper is novel to reformulate the spatio-temporal correspondence as contrastive random walk, which significantly enrichs self-supervisory signals that associates all paths and all patches across neighboring frames, while reduces compounding errors than existing methods that contrastively predict discriminative correspondences. - Numerous evaluations, ablation verifications and correspondence-driven applications make the claims sound. It is impressive that in may applications, the proposed self-supervised method not only beats previous self-supervised methods and even outperforms many carefully designed supervised methods.

Weaknesses: The proposed method is elegant with a solid theoretical grounding, it is simple yet effective. I am almost satisfied with the paper but has a question about the occlusion handling: occlusions are common between neighboring frames and they violate the cycle-consistency criterion. How does the proposed method deal with the occlusions? Will there be specific solutions to this issue apart from the proposed edge dropout?

Correctness: Yes. The claims, method, and the empirical methodoloy are correct.

Clarity: Yes. This paper is well written and pretty well organized.

Relation to Prior Work: Yes. This paper comprehensively discussed prior work and clearly states the differences from them.

Reproducibility: Yes

Additional Feedback:


Review 3

Summary and Contributions: Authors propose a self-supervised approach for learning a frame representation model capable of extracting correspondence over videos. They set up the problem as contrastive association between patch level features in adjacent frames. By cleverly converting videos into palindromes by concatenating a reversed video, they are able to supervise the representation by chaining the transition matrices between frames, and enforcing that each node ends up at itself at the end (using a contrastive objective across nodes). The learned representation is then evaluated for downstream tasks involving label propagation (eg DAVIS segmentation map propagation, JHMDB pose propagation etc), and authors are able to outperform prior self-supervised work by using a KNN approach over their learned representation.

Strengths: Soundness of claims [MEDIUM] ================ - The core approach is quite natural and explained well. The models, hyperparameters (inspired from moco-v1) used make sense for the tasks. - I especially liked the ablations in Fig 5, which show very consistent improvements with longer path lengths etc. - I also appreciate the comparison with other "standard" image level contrastive self supervised representations on the downstream tasks, like MOCO. This shows that the gains by contrasting patches over time leads to measurable gains. The discussion comparing the approach to standard contrastive learning in L112 is quite insightful. Significance and Novelty [MEDIUM] ================== - The core novelty of the paper is extending the idea of temporal cyclic consistency for learning correspondences (explored in multiple previous papers, perhaps most notably in [66]) to using a contrastive loss formulation and sidestepping the need for hand-crafted trackers. This idea is quite natural, and fits nicely in the contrastive formulation, which has been quite successful in representation learning more generally (MOCO, SimCLR, PIRL). - The results are decently better compared to prior work (2-4% on most benchmarks/evaluation settings compared to SOTA self-supervised representations). Hence, I think the contributions are decently significant. Relevance to NeurIPS [HIGH] ================ - The paper is definitely very relevant to the NeurIPS community as it explores self-supervised representation learning, and shows improved representations for interesting downstream label propagation tasks on standard benchmarks.

Weaknesses: Soundness of claims [MEDIUM] ================ 1. Analysis of this representation beyond video tasks As authors also discuss in L112, their approach can be thought of as a generalization of image-level representation learning with latent views. Moreover, given the fact authors are learning a frame level representation (as opposed to using 3D convolutions for videos etc), the representation is essentially image-level. it raises the question, how does it perform on standard image tasks, say imagenet linear classification? While authors do compare MOCO on their downstream tasks, I think the reverse would be interesting as well. While I do understand that since authors' model is pre-trained on kinetics so may not do as well compared to approaches like MOCO/SimCLR that are trained on ImageNet, in principle authors' method can be trained on image-level datasets as well, as per the approach they describe in L112. I think further experimental evaluation of those claims will make the paper quite a bit stronger. 2. How many nodes/patches do you need at training time? Authors fix the number of patches used at training time (49). Given that reliance of contrastive approaches on having a good set of negatives, I wonder how the performance varies with the granularity at which these patches are extracted? Perhaps an ablation comparing different number of these nodes would be quite informative. 3. In the ablations, authors find improvements using edge dropout, longer path training, test time adaptation, however the final results reported for all tasks are without those. Is there a reason to not use the best model/techniques for the final numbers on all the tasks? I feel this makes the paper a bit confusing for future readers, who may just pick the results out of Table 1 and 2, and ignore the improved results shown in Fig 5.

Correctness: To my understanding all comparison is done in the standard setup used in self-supervised label propagation using k-NN at test time (as also used in [32,33,66]). Hence the numbers etc should be comparable to prior work.

Clarity: The paper is generally well written and easy to read. Figures are helpful in better understanding the approach. However, one portion of the paper that was somewhat confusing to me is how the node features are extracted at test time. While at training time authors only consider 49 nodes per frame, at test time they are solving a pixel-level correspondence task so they likely need a lot more nodes per frame? They do mention that they are using a convolutional feature map but that will still downsample the feature map quite a bit. I would appreciate if the authors can clarify in the paper and rebuttal.

Relation to Prior Work: The paper has an extensive section which covers relevant related work in comparison to theirs. Many of the relevant approaches are also compared against in the experiments. One missing reference they should perhaps include is: Temporal Cycle-Consistency Learning (Dwibedi et al. CVPR'19). While it doesn't tackle label propagation as explored in this paper, the core idea of using temporal consistency in videos as a supervisory signal is quite relevant.

Reproducibility: Yes

Additional Feedback: - Algorithm 1 typos/suggestions: - v = l2_norm(linear(resnet(x))) - A = bmm(v[:,:,:-1], v[:,:,1:]), and shouldn't the output dimensions be B x (T-1) x P x P? If so the random walk loop should run over 2T-2 transitions between the 2T frames. - I noticed that a lot of hyperparameters mentioned in the approach section are based off of MOCO-v1. Authors could perhaps also try parameters from MOCO-v2/SimCLR (eg higher values of temperature ~0.1 instead of 0.07, MLP instead of linear layer for projection etc), which might help get further improvement in numbers for the final version of the paper. ------ Final Rating I have looked through the rebuttal and other reviews. I appreciate authors efforts in clarifying some of the details and responding to reviewers concerns. I plan to stick to my original rating.


Review 4

Summary and Contributions: The paper presented a self-supervised learning approach for learning long range space-time correspondence in video. The core idea is to leverage cycle consistency by considering a random walker on a video graph. Specifically, a video is represented as a space-time graph with nodes as patches and edges as affinity in feature space. A random walker on this graph is likely to return to its source patch given a palindrome video. This is encoded into a supervisory signal for learning the edge weights on the graph, leading to the feature representation of patches and the correspondence between patches. The idea is simple and elegant. The authors also present extensive experiments across several tasks using the learned representation for patch matching. The experiments are solid and the results are impressive.

Strengths: * The paper develops self-supervised learning for video correspondence. Both are important topics in machine learning and computer vision. * The perspective of learning to walk on a space-time graph is very interesting and quite novel. * The experiments are well-thought-of and the empirical results are strong. Overall, the paper has a good idea, major technical innovations (in terms of modeling) and strong results. I found the paper quite exciting and would advocate for its acceptance.

Weaknesses: I did not identify any major weakness of the paper.

Correctness: The claims, the method and the experiments are sound.

Clarity: The paper is well written. The intuition and the technical details are clearly explained. I enjoyed reading this paper.

Relation to Prior Work: Relation to prior work was sufficiently discussed in the paper.

Reproducibility: Yes

Additional Feedback: I am curious about how the method can learn to cheat (P4 footnote) and would like to see a bit more description / discussion here. What does the shortcut look like (mentioned in the supplementary material)? It seems that the model can learn to remember the spatial location of a patch, thus making the walker stay at the same location. **Post Rebuttal Update: The rebuttal has addressed my previous question on the shortcut. Overall, I strongly recommend the acceptance of this paper.

[Author Response · NeurIPS 2020]

We thank the reviewers for their thoughtful feedback and effort. It is a pleasure to receive such meaningful reviews.

**Texture-less Patches (R1).** Our model handles ambiguity by forming "soft" matches: in contrast to previous work [5], it considers a distribution over possible matches. Moreover, we can show that when a match is fundamentally ambiguous, the gradients of the loss are not detrimental because they "cancel": e.g., when a set of potential matches are identical (e.g. texture-less patches of the sky), the matching distribution will be perfectly uniform over this set, and the sum of gradients they contribute will be zero in the worst case. We will provide a proof in an updated version.

**Partial Occlusions (R1, R2).** The soft matching formulation is also robust to partial occlusion; by considering a distribution over matches for each node, it can form many-to-one and one-to-many matches, as well as matches at a region-level. Moreover, edge dropout (as well as spatial jittering) at training-time simulate similar challenges, providing further robustness, as can be seen in the improvement in J-recall in Table 1.

**Object Moves out of Camera (R1, R2).** Indeed, object disappearances do occur in less curated videos, such as those of our training set, Kinetics. The fact that our model learns a useful representation despite their presence suggests that it is robust to such cases. We hypothesize this is because total occlusions are infrequent and thus contribute only occasional label noise into the soft matching problem. Moreover, the use of sub-cycles (L148) also allows for learning from sub-sequences that may not be affected by total occlusions. Interestingly, we found that skip connections in time used to address occlusion in previous work [5] to not be beneficial. Incorporating additional temporal context in the right way is an important direction for future work.

**Shortcut Solutions (R1, R2, R4).** As the reviewers point out, learning from raw video involves the challenges mentioned above, such that it may be easier to learn a shortcut solution than model pixel appearance. As we mentioned in Footnote 1 ( L145) and Appendix C (L9), we observed that naively using a single spatial feature map to obtain node embeddings at training time can lead to the network solving the matching objective without using visual appearance. This is evidenced by a solution obtaining very low training error (despite a large training dataset), even when asked to match high-resolution maps, while transferring poorly to downstream label propagation tasks. It can also be qualitatively observed in visualizations of feature maps, which we will add to the Appendix. Even after reducing boundary effects (which have been shown to be useful for encoding positional information [3]) by considering architectures without padding, as well as mitigating leakage of global information through batch normalization [2], we found that a shortcut could still be learned. Ultimately, we adopt a simple solution used in prior work [1, 4]: removing relative position cues altogether, by cropping the image to obtain patches. This works because predicting relative position of crops is challenging [1, 4]. This leads to meaningful modeling of visual appearance, as indicated in our evaluation experiments.

**Feature Extraction (R1, R3).** We extract separate patches at training time. At test-time, we can efficiently compute node embeddings by creating a single feature map and then flattening the spatial dimensions. We follow previous work [5]: our encoder makes a feature map 1/8 the the image resolution (e.g. $112 \times 60$ for $900 \times 480$px test images), and the label map is up-sampled with nearest-neighbor interpolation for evaluation.

**Contrastive Learning with "Latent" Views (R1).** By "latent" view, we mean that the corresponding positive view for each query is not known, as is the case in learning correspondence without labels; this is in contrast to typical contrastive learning settings, in which positive views are generated by hand-crafted data augmentation strategies. As scenes changes between frames, it is rare that corresponding nodes have exactly the same representation; it is invariance to spatio-temporal changes, rather than semantics, that we seek to learn. In our work, we leverage a path-level constraint (cycle-consistency) to guide the inference of latent views. We will clarify this in an updated version.

**Image Classification Tasks (R3).** Our main interest is a sense of similarity useful for finding space-time correspondence. While approaches like MoCo [2], seek invariance across instances of the same class, our representation distinctly should not. We view these two aims to play complementary roles in the spectrum of information needed to model instances and categories, in that we capture the natural data augmentation exhibited in dynamic scenes. More technically, unlike methods that rely on negatives drawn from other examples, our negatives come from within instance itself, so as to distinguish its parts from one another. That said, it is definitely interesting to consider how spatio-temporal invariance can give rise to semantic representations, and we will investigate image classification in an updated version.

**Extending Related Work (R1, R3).** We completely agree that connections to recent work on attention-based architectures for encoding sets and graphs, as well as the work of Dwibedi et al, are interesting and deserve discussion. If accepted, we look forward to using the extra page in the camera-ready to do so.

**Clarification in Pseudocode (R3).** We appreciate that reviewers took the time to read our pseudocode, and hope it was of help. Indeed, the output dimension of that operation is `B x (T-1) x P x P`. We will clarify this.

**Training Hyper-parameters (R1, R3).** While our hyper-parameters (Appendix E) follow prior work [2, 4] due to computational constraints, we agree that leveraging more recent hyper-parameters might result in further improvements.

[1] C Doersch et al. Unsupervised visual representation learning by context prediction. In *ICCV*, 2015.
[2] K He et al. Momentum contrast for unsupervised visual representation learning. In *CVPR*, 2020.
[3] M Islam et al. How much position information do convolutional neural networks encode? *ICLR*, 2020.
[4] A Oord et al. Representation learning with contrastive predictive coding. *arXiv:1807.03748*, 2018.
[5] X Wang et al. Learning correspondence from the cycle-consistency of time. In *CVPR*, 2019.


[Meta-Review · NeurIPS 2020]

The reviewers all agreed that this work tackles an important topic in computer vision and the proposed solution is elegant. They also praised that numerous evaluations/ablations and strong results make the proposed approach convincing.